# Recurrent linear models of simultaneously-recorded neural populations

**Marius Pachitariu, Biljana Petreska, Maneesh Sahani**
Gatsby Computational Neuroscience Unit
University College London, UK
{marius,biljana,maneesh}@gatsby.ucl.ac.uk

## Abstract

Population neural recordings with long-range temporal structure are often best understood in terms of a common underlying low-dimensional dynamical process. Advances in recording technology provide access to an ever-larger fraction of the population, but the standard computational approaches available to identify the collective dynamics scale poorly with the size of the dataset. We describe a new, scalable approach to discovering low-dimensional dynamics that underlie simultaneously recorded spike trains from a neural population. We formulate the Recurrent Linear Model (RLM) by generalising the Kalman-filter-based likelihood calculation for latent linear dynamical systems to incorporate a generalised-linear observation process. We show that RLMs describe motor-cortical population data better than either directly-coupled generalised-linear models or latent linear dynamical system models with generalised-linear observations. We also introduce the cascaded generalised-linear model (CGLM) to capture low-dimensional instantaneous correlations in neural populations. The CGLM describes the cortical recordings better than either Ising or Gaussian models and, like the RLM, can be fit exactly and quickly. The CGLM can also be seen as a generalisation of a low-rank Gaussian model, in this case factor analysis. The computational tractability of the RLM and CGLM allow both to scale to very high-dimensional neural data.

## 1 Introduction

Many essential neural computations are implemented by large populations of neurons working in concert, and recent studies have sought both to monitor increasingly large groups of neurons [1, 2] and to characterise their collective behaviour [3, 4]. In this paper we introduce a new computational tool to model coordinated behaviour in very large neural data sets. While we explicitly discuss only multi-electrode extracellular recordings, the same model can be readily used to characterise 2-photon calcium-marker image data, EEG, fMRI or even large-scale biologically-faithful simulations.

Populational neural data may be represented at each time point by a vector $y_t$ with as many dimensions as neurons, and as many indices $t$ as time points in the experiment. For spiking neurons, $y_t$ will have positive integer elements corresponding to the number of spikes fired by each neuron in the time interval corresponding to the $t$-th bin. As others have before [5, 6], we assume that the coordinated activity reflected in the measurement $y_t$ arises from a low-dimensional set of processes, collected into a vector $x_t$, which is not directly observed. However, unlike the previous studies, we construct a recurrent model in which the hidden processes $x_t$ are driven directly and explicitly by the measured neural signals $y_1 \ldots y_{t-1}$. This assumption simplifies the estimation process. We assume for simplicity that $x_t$ evolves with linear dynamics and affects the future state of the neural signal $y_t$ in a generalised-linear manner, although both assumptions may be relaxed. As in the latent dynamical system, the resulting model enforces a "bottleneck", whereby predictions of $y_t$ based on $y_1 \ldots y_{t-1}$ must be carried by the low-dimensional $x_t$.

State prediction in the RLM is related to the Kalman filter [7] and we show in the next section a formal equivalence between the likelihoods of the RLM and the latent dynamical model when observation noise is Gaussian distributed. However, spiking data is not well modelled as Gaussian, and the generalisation of our approach to Poisson noise leads to a departure from the latent dynamical approach. Unlike latent linear models with conditionally Poisson observations, the parameters of our model can be estimated efficiently and without approximation. We show that, perhaps in consequence, the RLM can provide superior descriptions of neural population data.

## 2 From the Kalman filter to the recurrent linear model (RLM)

Consider a latent linear dynamical system (LDS) model with linear-Gaussian observations. Its graphical model is shown in Fig. 1A. The latent process is parametrised by a dynamics matrix $A$ and innovations covariance $Q$ that describe the evolution of the latent state $\boldsymbol{x}_t$:

$$P(\boldsymbol{x}_t|\boldsymbol{x}_{t-1}) = \mathcal{N}(\boldsymbol{x}_t|A\boldsymbol{x}_{t-1}, Q),$$

where $\mathcal{N}(x|\mu, \Sigma)$ represents a normal distribution on $x$ with mean $\mu$ and (co)variance $\Sigma$. For brevity, we omit here and below the special case of the first time-step, in which $\boldsymbol{x}_1$ is drawn from a multivariate Gaussian. The output distribution is determined by an observation loading matrix $C$ and a noise covariance $R$ often taken to be diagonal so that all covariance is modelled by the latent process:

$$P(\boldsymbol{y}_t|\boldsymbol{x}_t) = \mathcal{N}(\boldsymbol{y}_t|C\boldsymbol{x}_t, R).$$

In the LDS, the joint likelihood of the observations $\{\boldsymbol{y}_t\}$ can be written as the product:

$$P(\boldsymbol{y}_1 \dots \boldsymbol{y}_T) = P(\boldsymbol{y}_1) \prod_{t=2}^{T} P(\boldsymbol{y}_t|\boldsymbol{y}_1 \dots \boldsymbol{y}_{t-1})$$

and in the Gaussian case can be computed using the usual Kalman filter approach to find the conditional distributon at time $t$ iteratively:

$$
\begin{aligned}
P(\boldsymbol{y}_{t+1}|\boldsymbol{y}_1 \dots \boldsymbol{y}_t) &= \int d\boldsymbol{x}_{t+1} \ P(\boldsymbol{y}_{t+1}|\boldsymbol{x}_{t+1})P(\boldsymbol{x}_{t+1}|\boldsymbol{y}_1 \dots \boldsymbol{y}_t) \\
&= \int d\boldsymbol{x}_{t+1} \ \mathcal{N}(\boldsymbol{y}_{t+1}|C\boldsymbol{x}_{t+1}, R) \ \mathcal{N}(\boldsymbol{x}_{t+1}|A\hat{\boldsymbol{x}}_t, V_{t+1}) \\
&= \mathcal{N}(\boldsymbol{y}_{t+1}|CA\hat{\boldsymbol{x}}_t, CV_{t+1}C^\top + R),
\end{aligned}
$$

where we have introduced the (filtered) state estimate $\hat{\boldsymbol{x}}_t = \mathsf{E}\left[\boldsymbol{x}_t|\boldsymbol{y}_1 \dots \boldsymbol{y}_t\right]$ and (predictive) uncertainty $V_{t+1} = \mathsf{E}\left[(\boldsymbol{x}_{t+1} - A\hat{\boldsymbol{x}}_t)^2|\boldsymbol{y}_1 \dots \boldsymbol{y}_t\right]$. Both quantities are computed recursively using the Kalman gain $K_t = V_t C^\top (CV_t C^\top + R)^{-1}$, giving the following recursive recipe to calculate the conditional likelihood of $\boldsymbol{y}_{t+1}$:

$$
\begin{aligned}
\hat{\boldsymbol{x}}_t &= A\hat{\boldsymbol{x}}_{t-1} + K_t(\boldsymbol{y}_t - \hat{\boldsymbol{y}}_t) \\
V_{t+1} &= A(I - K_t C)V_t A^\top + Q \\
\hat{\boldsymbol{y}}_{t+1} &= CA\hat{\boldsymbol{x}}_t \\
P(\boldsymbol{y}_{t+1}|\boldsymbol{y}_1 \dots \boldsymbol{y}_t) &= \mathcal{N}(\boldsymbol{y}_{t+1}|\hat{\boldsymbol{y}}_{t+1}, CV_{t+1}C^\top + R)
\end{aligned}
$$

For the Gaussian LDS, the Kalman gain $K_t$ and state uncertainty $V_{t+1}$ (and thus the output covariance $CV_{t+1}C^\top + R$) depend on the model parameters $(A, C, R, Q)$ and on the time step—although as time grows they both converge to stationary values. Neither depends on the observations.

Thus, we might consider a relaxation of the Gaussian LDS model in which these matrices are taken to be stationary from the outset, and are parametrised independently so that they are no longer constrained to take on the "correct" values as computed for Kalman inference. Let us call this parametric form of the Kalman gain $W$ and the parametric form of the output covariance $S$. Then the conditional likelihood iteration becomes

$$
\begin{aligned}
\hat{\boldsymbol{x}}_t &= A\hat{\boldsymbol{x}}_{t-1} + W(\boldsymbol{y}_t - \hat{\boldsymbol{y}}_t) \\
\hat{\boldsymbol{y}}_{t+1} &= CA\hat{\boldsymbol{x}}_t \\
P(\boldsymbol{y}_{t+1}|\boldsymbol{y}_1 \dots \boldsymbol{y}_t) &= \mathcal{N}(\boldsymbol{y}_{t+1}|\hat{\boldsymbol{y}}_{t+1}, S).
\end{aligned}
$$

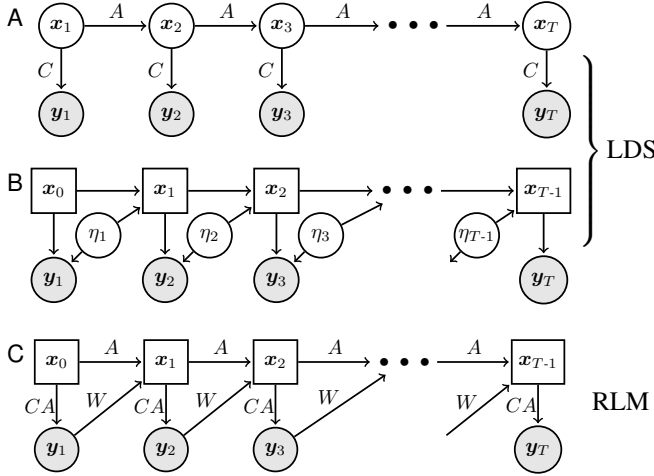

Figure 1: Graphical representations of the latent linear dynamical system (LDS: A, B) and recurrent linear model (RLM: C). Shaded variables are observed, unshaded circles are latent random variables and squares are variables that depend deterministically on their parents. In B the LDS is redrawn in terms of the random innovations $\eta_t = \boldsymbol{x}_t - A\boldsymbol{x}_{t-1}$, facilitating the transition towards the RLM. The RLM is then obtained by replacing $\eta_t$ with a deterministically derived estimate $W(\boldsymbol{y}_t - \hat{\boldsymbol{y}}_t)$.

The parameters of this new model are $A, C, W$ and $S$. This is a relaxation of the Gaussian latent LDS model because $W$ has more degrees of freedom than $Q$, as does $S$ than $R$ (at least if $R$ is constrained to be diagonal). The new model has a recurrent linear structure in that the random observation $\boldsymbol{y}_t$ is fed back linearly to perturb the otherwise deterministic evolution of the state $\hat{\boldsymbol{x}}_t$.

A graphical representation of this model is shown in Fig. 1C, along with a redrawn graph of the LDS model. The RLM can be viewed as replacing the random innovation variables $\eta_t = \boldsymbol{x}_t - A\boldsymbol{x}_{t-1}$ with data-derived estimates $W(\boldsymbol{y}_t - \hat{\boldsymbol{y}}_t)$; estimates which are made possible by the fact that $\eta_t$ contributes to the variability of $\boldsymbol{y}_t$ around $\hat{\boldsymbol{y}}_t$.

## 3 Recurrent linear models with Poisson observations

The discussion above has transformed a stochastic-latent LDS model with Gaussian output to an RLM with deterministic latent, but still with Gaussian output. Our goal, however, is to fit a model with an output distribution better suited to the binned point-processes that characterise neural spiking. Both linear Kalman-filtering steps above and the eventual stationarity of the inference parameters depend on the joint Gaussian structure of the assumed LDS model. They would not apply if we were to begin a similar derivation from an LDS with Poisson output. However, a tractable approach to modelling point-process data with low-dimensional temporal structure may be provided by introducing a generalised-linear output stage *directly* to the RLM. This model is given by:

$$\hat{\boldsymbol{x}}_t = A\hat{\boldsymbol{x}}_{t-1} + W(\boldsymbol{y}_t - \hat{\boldsymbol{y}}_t)$$
$$g(\hat{\boldsymbol{y}}_{t+1}) = CA\hat{\boldsymbol{x}}_t \qquad (1)$$
$$P(\boldsymbol{y}_{t+1}|\boldsymbol{y}_1 \dots \boldsymbol{y}_t) = \mathsf{ExpFam}(\boldsymbol{y}_{t+1}|\hat{\boldsymbol{y}}_{t+1})$$

where ExpFam is an exponential-family distribution such as Poisson, and the element-wise link function $g$ allows for a nonlinear mapping from $\boldsymbol{x}_t$ to the predicted mean $\hat{\boldsymbol{y}}_{t+1}$. In the following, we will write f for the inverse-link as is more common for neural models, so that $\hat{\boldsymbol{y}}_{t+1} = \mathrm{f}(CA\hat{\boldsymbol{x}}_t)$.

The simplest Poisson-based generalised-linear RLM might take as its output distribution

$$P(\boldsymbol{y}_t|\hat{\boldsymbol{y}}_t) = \prod_i \mathsf{Poisson}(y_{ti}|\hat{y}_{ti}); \qquad \hat{\boldsymbol{y}}_t = \mathrm{f}(CA\hat{\boldsymbol{x}}_{t-1})),$$

where $y_{ti}$ is the spike count of the $i$th cell in bin $t$ and the function $f$ is non-negative. However, comparison with the output distribution derived for the Gaussian RLM suggests that this choice would fail to capture the instantaneous covariance that the LDS formulation transfers to the output distribution (and which appears in the low-rank structure of $S$ above). We can address this concern in two ways. One option is to bin the data more finely, thus diminishing the influence of the instantaneous covariance. The alternative is to replace the independent Poissons with a correlated output distribution on spike counts. The cascaded generalised-linear model introduced below is a natural choice, and we will show that it captures instantaneous correlations faithfully with very few hidden dimensions.

In practice, we also sometimes add a fixed input $\boldsymbol{\mu}_t$ to equation 1 that varies in time and determines the average behavior of the population or the peri-stimulus time histogram (PSTH).

$$\hat{\boldsymbol{y}}_{t+1} = \mathrm{f}\left(\boldsymbol{\mu}_t + CA\boldsymbol{x}_t\right)$$

Note that the matrices $A$ and $C$ retain their interpretation from the LDS models. The matrix $A$ controls the evolution of the dynamical process $\boldsymbol{x}_t$. The phenomenology of its dynamics is determined by the complex eigenvalues of $A$. Eigenvalues with moduli close to 1 correspond to long timescales of fluctuation around the PSTH. Eigenvalues with non-zero imaginary part correspond to oscillatory components. Finally, the dynamics will be stable iff all the eigenvalues lie within the unit disc. The matrix $C$ describes the dependence of the high-dimensional neural signals on the low-dimensional latent processes $\boldsymbol{x}_t$. In particular, equation 2 determines the firing rate of the neurons. This generalised-linear stage ensures that the firing rates are positive through the link function f, and the observation process is Poisson. For other types of data, the generalised-linear stage might be replaced by other appropriate link functions and output distributions.

### 3.1 Relationship to other models

RLMs are related to recurrent neural networks [8]. The differences lie in the state evolution, which in the neural network is nonlinear: $\boldsymbol{x}_t = h\left(A\boldsymbol{x}_{t-1} + W\boldsymbol{y}_{t-1}\right)$; and in the recurrent term which depends on the observation rather than the prediction error. On the data considered here, we found that using sigmoidal or threshold-linear functions $h$ resulted in models comparable in likelihood to the RLM, and so we restricted our attention to simple linear dynamics. We also found that using the prediction error term $W\left(\boldsymbol{y}_{t-1} - \hat{\boldsymbol{y}}_t\right)$ resulted in better models than the simple neural-net formulation, and we attribute this difference to the link between the RLM and Kalman inference.

It is also possible to work within the stochatic latent LDS framework, replacing the Gaussian output distribution with a generalised-linear Poisson output (e.g. [6]). The main difficulty here is the intractability of the estimation procedure. For an unobserved latent process $\boldsymbol{x}_t$, an inference procedure needs to be devised to estimate the posterior distribution on the entire sequence $\boldsymbol{x}_1 \ldots \boldsymbol{x}_t$. For linear-Gaussian observations, this inference is tractable and is provided by Kalman smoothing. However, with generalised-linear observations, inference becomes intractable and the necessary approximations [6] are computationally intense and can jeopardize the quality of the fitted models. By contrast, in the RLM $\boldsymbol{x}_t$ is a deterministic function of data. In effect, the Kalman filter has been built into the model as the accurate estimation procedure, and efficient fitting is possible by direct gradient ascent on the log-likelihood. Empirically we did not encounter difficulties with local minima during optimization, as has been reported for LDS models fit by approximate EM [9]. Multiple restarts from different random values of the parameters always led to models with similar likelihoods.

Note that to estimate the matrices $A$ and $W$ the gradient must be backpropagated through successive iterations of equation 1. This technique, known as backpropagation-through-time, was first described by [10] as a technique to fit recurrent neural network models. Recent implementations have demonstrated state-of-the-art language models [11]. Backpropagation-through-time is thought to be inherently unstable when propagated past many timesteps and often the gradient is truncated prematurely [11]. We found that using large values of momentum in the gradient ascent alleviated these instabilities and allowed us to use backpropagation without the truncation.

## 4 The cascaded generalised-linear model (CGLM)

The link between the RLM and the LDS raises the possibility that a model for simultaneously-recorded correlated spike counts might be derived in a similar way, starting from a non-dynamical, but low-dimensional, Gaussian model. Stationary models of population activity have attracted recent interest for their own sake (e.g. [1]), and would also provide a way model correlations introduced by common innovations that were neglected by the simple Poisson form of the RLM. Thus, we consider vectors $\boldsymbol{y}$ of spike counts from $N$ neurons, without explicit reference to the time at which they were collected. A Gaussian model for $\boldsymbol{y}$ can certainly describe correlations between the cells, but is ill-matched to discrete count observations. Thus, as with the derivation of the RLM from the Kalman filter, we derive here a new generalisation of a low-dimensional, structured Gaussian model to spike count data.

The distribution of any multivariate variable $\boldsymbol{y}$ can be factorized into a "cascaded" product of multiple one-dimensional distributions:

$$\mathrm{P}\left(\boldsymbol{y}\right) = \prod_{n=1}^{N} \mathrm{P}\left(y_n | \boldsymbol{y}_{<n}\right). \tag{2}$$

Here $n$ indexes the neurons up to the last neuron $N$, and $\boldsymbol{y}_{<n}$ is the $(n$–$1)$-vector $[y_1 \ldots y_{n-1}]$. For a Gaussian-distributed $\boldsymbol{y}$, the conditionals $\mathrm{P}\left(y_n | \boldsymbol{y}_{<n}\right)$ would be linear-Gaussian. Thus, we propose the "cascaded generalised linear model" (CGLM) in which each such one-dimensional conditional distribution is a generalised-linear model:

$$\hat{y}_n = \mathrm{f}\left(\mu_n + S_n^T \boldsymbol{y}_{<n}\right) \tag{3}$$

$$\mathrm{P}\left(y_n | \boldsymbol{y}_{<n}\right) = \mathsf{ExpFam}\left(\hat{y}_n\right) \tag{4}$$

and in which the linear weights $S_n$ take on a structured form developed below.

The equations 3 and 4 subsume the Gaussian distribution with arbitrary covariance in the case that f is linear, and the $\mathsf{ExpFam}$ conditionals are Gaussian. In this case, for a joint covariance of $\Sigma$, it is straightforward to derive the expression

$$S_n = \frac{1}{\left(\Sigma_{\le n, \le n}\right)_{n,n}^{-1}} \left(\Sigma_{\le n, \le n}\right)_{n,<n}^{-1}. \tag{5}$$

where the subscripts $<n$ and $\le n$ restrict the matrix to the first $(n-1)$ and $n$ rows and/or columns respectively. Thus, we might construct suitably structured linear weights for the CGLM by applying this result to the covariance matrix induced by the low-dimensional Gaussian model known as factor analysis [12]. Factor analysis assumes that data are generated from a $K$-dimensional latent process $\boldsymbol{x} \sim \mathcal{N}\left(0, I\right)$, where $I$ is the $K \times K$ identity matrix, and $\boldsymbol{y}$ has the conditional distribution $\mathrm{P}\left(\boldsymbol{y}|\boldsymbol{x}\right) = \mathcal{N}\left(\Lambda \boldsymbol{x}, \Psi\right)$ with $\Psi$ a diagonal matrix and $\Lambda$ an $N \times K$ loading matrix. This leads to a covariance of $\boldsymbol{y}$ given by $\Sigma = \Psi + \Lambda \Lambda^T$. If we repeat the derivation of equations 3, 4 and 5 for this covariance matrix, we obtain an expression for $S_n$ via the matrix inversion lemma:

$$\begin{aligned}
S_n &= \frac{1}{\left(\Sigma_{\le n, \le n}\right)_{n,n}^{-1}} \left(\Psi_{\le n, \le n} + \Lambda_{\le n,\cdot} \Lambda_{\le n,\cdot}^T\right)_{n,<n}^{-1} \\
&= \frac{1}{\left(\Sigma_{\le n, \le n}\right)_{n,n}^{-1}} \left(\Psi_{\le n, \le n}^{-1} - \Psi_{\le n, \le n}^{-1} \Lambda_{<n,\cdot} \left(\cdots\right) \Lambda_{<n,\cdot}^T \Psi_{<n,<n}^{-1}\right)_{n,<n} \\
&= -\frac{1}{\left(\Sigma_{\le n, \le n}\right)_{n,n}^{-1}} \left(\left(\Psi^{-1}\Lambda\right)_{\le n,\cdot} \left(\cdots\right) \left(\Lambda \Psi^{-1}\right)_{\le n,\cdot}^T\right)_{n,<n}
\end{aligned} \tag{6}$$

where the omitted factor $(\cdots)$ is a $K \times K$ matrix. The first term in equation 6 vanishes because it involves only the off-diagonal entries of $\Psi$. The surviving factor shows that $S_n$ is formed by taking a linear combination of the columns of $\Psi^{-1}\Lambda$ and then truncating to the first $n-1$ elements. Thus, if we arrange all $S_n$ as the upper columns of an $N \times N$ matrix $S$, we can write $S = \mathrm{upper}\left(zw^T\right)$ for some low-dimensional matrices $z = \Psi^{-1}\Lambda$ and $w$, where the operation $\mathrm{upper}$ extracts the strictly upper triangular part of a matrix. This is the natural structure imposed on the cascaded conditionals by factor analysis. Thus, we adopt the same constraint on $S$ in the case of generalised-linear observations. The resulting (CGLM) is shown below to provide better fits to binarized neural data than standard Ising models (see the Results section), even with as few as three latent dimensions.

Another useful property of the CGLM is that it allows stimulus-dependent inputs in equation 3. The CGLM can also be used in combination with the generalised-linear RLM, with the CGLM replacing the otherwise independent observation model. This approach can be useful when large bins are used to discretize spike trains. In both cases the model can be estimated quickly with standard gradient ascent techniques.

## 5   Alternative models

### 5.1   Alternative for temporal interactions: causally-coupled generalised linear model

One popular and simple model of simultaneously recorded neuronal populations [3] constructs temporal dependencies between units by directly coupling each neuron's probability of firing to the past

spikes in the entire population:

$$\mathbf{y}_t \propto \mathrm{Poisson}(\mathrm{f}(\mu_t + \sum_{i=1}^{N} B_i \, (h_i \star \mathbf{y}_t)))$$

Here, $h_i \star \mathbf{y}_t$ are convolutions of the spike trains with a set of basis functions $h_i$, and $B_i$ are pairwise interaction weights. Each matrix $B_i$ has $N^2$ parameters where $N$ is the number of neurons, so the number of parameters grows quadratically with the population size. This type of scaling makes the model prohibitive to use with very large-scale array recordings. Even with aggressive regularization techniques, the model's parameters are difficult to identify with limited amounts of data. Perhaps more importantly, the model does not have a physical interpretation. Neurons recorded in cortex are rarely directly-connected and retinal ganglion cells almost never directly connect to each other. Instead, such directly-coupled GLMs are used to describe so-called 'functional' interactions between neurons [3]. We believe a much better interpretation for the correlations observed between pairs of neurons is that they are caused by common inputs to these neurons which seem often to be confined to a small number of dimensions. The models we propose here, the RLM and the CGLM, are aimed at discovering such inputs.

## 5.2 Alternative for instantaneous interactions: the Ising model

Instantaneous interactions between binary data (as would be obtained by counting spikes in short intervals) can be modelled in terms of their pairwise interactions [1] embodied in the Ising model:

$$\mathrm{P}\left(\boldsymbol{y}\right) = \frac{1}{Z} \; \mathrm{e}^{\boldsymbol{y}^T J \boldsymbol{y}}. \tag{7}$$

where $J$ is a pairwise interaction matrix and $Z$ is the partition function, or the normalization constant of the model. The model's attractiveness is that for a given covariance structure it makes the weakest possible assumptions about the distribution of $\boldsymbol{y}$, that is, like a Gaussian for continuous data, it has the largest possible entropy under the covariance constraint. However, the Ising model and the so-called functional interactions $J$ have no physical interpretation when applied to neural data. Furthermore, Ising models are difficult to fit as they require estimates of the gradients of the partition function $Z$; they also suffer from the same quadratic scaling in number of paramters as does the directly-coupled GLM. Ising models are even harder to estimate when stimulus-dependent inputs are added in equation 7, but for data collected in the retina or other sensory areas [1], much of the covariation in $\boldsymbol{y}$ may be expected to arise from common stimulus input. Another short-coming of the Ising model is that it can only model binarized data and cannot be normalized for integer $\boldsymbol{y}$-s [6], so either the time bins need to be reduced to ensure no neuron fires more than one spike in a single bin or the spike counts must be capped at 1.

## 6 Results

### 6.1 Simulated data

We began by evaluating RLM models fit to simulated data where the true generative parameters were known. Two aspects of the estimated models were of particular interest: the phenomenology of the dynamics (captured by the eigenvalues of the dynamics matrix $A$) and the relationship between the dynamical subspace and measured neural activity (captured by the output matrix $C$). We evaluated the agreement between the estimated and generative output matrices by measuring the principal angles between the corresponding subspaces. These report, in succession, the smallest angle achievable between a line in one subspace and a line in the second subspace, once all previous such vectors of maximal agreement have been projected out. Exactly aligned $n$-dimensional subspaces have all $n$ principal angles equal to $0°$. Unrelated low-dimensional subspaces embedded in high dimensions are close to orthogonal and so have principal angles near $90°$.

We first verified the robustness of maximisation of the generalised-linear RLM likelihood by fitting models to simulated data generated by a known RLM. Fig. 2(a) shows eigenvalues from several simulated RLMs and the eigenvalues recovered by fitting parameters to simulated data. The agreement is generally good. In particular, the qualitative aspects of the dynamics reflected in the absolute values and imaginary parts of the eigenvalues are well characterised. Fig. 2(d) shows that the RLM fits

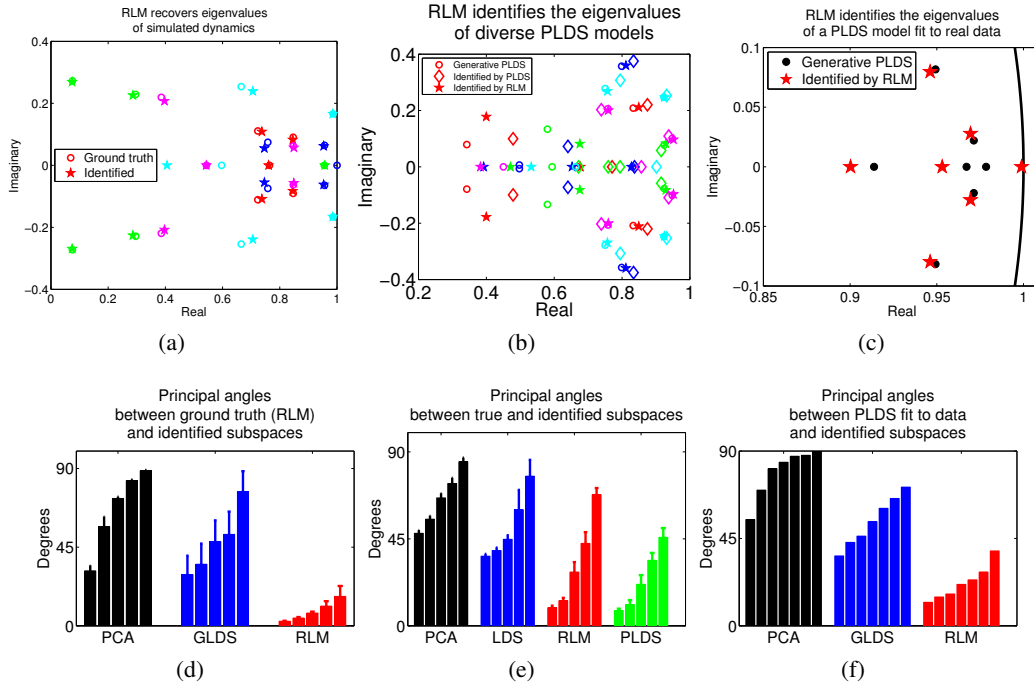

Figure 2: Experiments on 100-dimensional simulated data generated from a 5-dimensional latent process. Generating models were Poisson RLM (ad), Poisson LDS with random parameters (cf) and Poisson LDS model with parameters fit to neural data (cf). The models fit were PCA, LDS with Gaussian (LDS/GLDS) or Poisson (PLDS) output, and RLM with Poisson output (RLM). In the upper plots, eigenvalues from different runs are shown in different colors.

also recover the subspace defined by the loading matrix $C$, and do so substantially more accurately than either principal components analysis (PCA) or GLDS models. It is important to note that the likelihoods of LDS models with Poisson observations are difficult to optimise, and so may yield poor results even when fit to within-class data. In practice we did not observe local optima with the RLM or CGLM.

We also asked whether the RLM could recover the dynamical properties and latent subspace of data generated by a latent LDS model with Poisson observations. Fig. 2(b) shows that the dynamical eigenvalues of the maximum-likelihood RLM are close to the eigenvalues of generative LDS dynamics, whilst Fig. 2(e) shows that the dynamical subspace is also correctly recovered. Parameters for these simulations were chosen randomly. We then asked whether the quality of parameter identification extended to Poisson-output LDS models with realistic parameters, by generating data from a Poisson-output LDS model that had been fit to a neural recording. As seen in figs. 2(c) and 2(f), the RLM fits remain accurate in this regime, yielding better subspace estimates than either PCA or a Gaussian LDS.

## 6.2 Array recorded data

We next compared the performance of the novel models on neural data. The RLM was compared to the directed-coupled GLM (fit by gradient-based likelihood optimisation) as well as LDS models with Gaussian or Poisson outputs (fit by EM, with a Laplace approximation E-step). The CGLM was compared to the Ising model. We used a dataset of 92 neurons recorded with a Utah array implanted in the premotor and motor cortices of a rhesus macaque monkey performing a delayed center-out reach task. For all comparisons below we use datasets of 108 trials in which the monkey made movements to the same target.

We discretized spike trains into time bins of 10ms. The directed-coupled GLM needed substantial regularization in order to make good predictions on held-out test data. Figure 3(a) shows only the best cross-validation result for the GLM, but results without regularization for models with

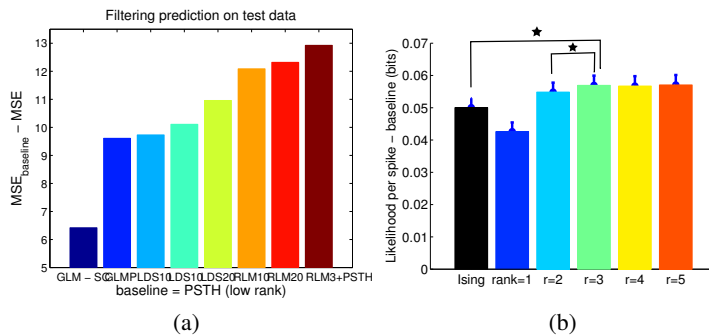

| (a) | (b) |

Figure 3: a. Predictive performance of various models on test data (higher is better). GLM-type models are helped greatly by self-coupling filters (which the other models do not have). The best model is an RLM with three latent dimensions and a low-rank model of the PSTH (see the supplementary material for more information about this model). Adding self-coupling filters to this model further increases its predictive performance by 5 (not shown). b. The likelihood per spike of Ising models as well as CGLM models with small numbers of hidden dimensions. The CGLM saturates at three dimensions and performs better than Ising models.

low-dimensional parametrisation. Performance was measured by the causal mean-squared-error in prediction subtracted from the error of a low-rank smoothed PSTH model (based on a singular-value decomposition of the matrix of all smoothed PSTHs). The number of dimensions (5) and the standard deviation of the Gaussian smoothing filter (20 ms) were cross-validated to find the best possible PSTH performance. Thus, our evaluation is focuses on each model's ability to predict trial-to-trial co-variation in firing around the mean.

A second measure of performance for the RLM was obtained by studying probabilistic samples obtained from the fitted model. Figure 4 in the supplemental material shows averaged noise cross-correlograms obtained from a large set of samples. Note that the PSTHs have been subtracted from each trial to reveal only the extra correlation structure that is not repeated amongst trials. Even with few hidden dimensions, the model captures well the full temporal structure of the noise correlations.

In the case of the Ising model we binarized the data by replacing all spike counts larger than 1 with 1. The log-likelihood of the Ising model could only be estimated for small numbers of neurons, so for comparison we took only the 30 most active neurons. The measure of performance reported in figure 3(b) is the extra log-likelihood per spike obtained above that of a model that makes constant predictions equal to the mean firing rate of each neuron. The CGLM model with only three hidden dimensions achieves the best generalisation performance, surpassing the Ising model. Similar results for the performance of the CGLM can be seen on the full dataset of 92 neurons with non-binarized data, indicating that three latent dimensions suffice to describe the full space visited by the neuronal population on a trial-by-trial basis.

# 7 Discussion

The generalised-linear RLM model, while sharing motivation with latent LDS model, can be fit more efficiently and without approximation to non-Gaussian data. We have shown improved performance on both simulated data and on population recordings from the motor cortex of behaving monkeys.

The model is easily extended to other output distributions (such as Bernoulli or negative binomial), to mixed continuous and discrete data, to nonlinear outputs, and to nonlinear dynamics. For the motor data considered here, the generalised-linear model performed as well as models with further non-linearites. However, preliminary results on data from sensory cortical areas suggests that non-linear models may be of greater value in other settings.

# 8 Acknowledgments

We thank Krishna Shenoy and members of his lab for generously providing access to data. Funding from the Gatsby Charitable Foundation and DARPA REPAIR N66001-10-C-2010.

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
