[Supplementary Material]



Figure 4: Samples from an RLM model resemble the cross-correlation structure of the data. Neurons were seprated into four groups ordered by their total correlation and the average cross-correlograms within each group are shown. Continuous lines are the data and the model generated samples are in dashed lines.

Figure 5: When a low-dimensional PSTH model ($\mu$) is added to the RLM, log-likelihood saturates at a very small number of latent dimensions (three) and performs better than RLM and LDS models without PSTH terms. Without the PSTH model, we needed to use as many as ten or twenty latent dimensions to capture the full aspects of the data. Figure b) shows the magnitudes of the eigenvalues obtained with RLM3+PSTH. Most of the trial-by-trial variability is thus explained with timescales of 100 and 200 ms.