[Reviews · NeurIPS 2013]

Submitted by Assigned_Reviewer_1

The authors present a new descriptive model of spiking dynamics recorded in neural populations. The authors claim that the major advantage of the new model is that it can be fitted easily and more directly to experimental data than previous models and therefore it has a better performance.

The paper is generally well written, although I found that the Results section is less clear than the presentation of the models. Details about the training and testing procedure are missing. The work is original and it makes an interesting progress in the models of neural population data. Testing of the model against its alternatives is not convincing in the present form and it needs to be clarified.

The authors use many acronyms for different models and it is not always clear which one they are referring to. A tabular or a graphical (tree-like) representation showing the similarities and differences between these models could also be useful. (RLM, RNN, CLM, LDS, GLDS, PLDS, gl-RLM, GLM)

Specific comments:
line 134: All GLDS have an equivalent GRLM, but not all GRLM is equivalent to a GLDS. Putting this in a different way, these parameters are not independent, W and S depends on C. Similarly, is it true that a PLDS is a special case of RLMs with Poisson observation? So later in the Results al Data generated by PLDS could be, in principle, recovered by the appropriate class of RLMs?

line 172: The authors describe two ways to handle instantaneous correlations in RLMs. It is not always clear which one is used later in the Results section.
line 193: W \hat{y}_t = W f(CA\hat{x}_t) that could be (approximately) incorporated into the hidden dynamics.
line 197: The authors use the acronym PLDS for the Poisson LDS model later.
line 235: vectors are usually typed in boldface in the paper, but I got completely lost here. y_n is a scalar, y_{ < n} is a vector but the typeset often suggests the opposite!
line 238, Eq. 3: is \mu_n and \mu_{ < n} = 0 ? If not, isn't it missing from (y_{ < n} - \mu_{ < n})?
line 245: I do not see why we have (\Sigma_{ < n, < n})^{-1} instead of (\Sigma)^{-1} (same for line 255)
line 257: I could not find how this line follows from the previous one and from the matrix inversion lemma, and I could not estimate how much we loose on the approximation.
line 270: Alternative models. Does this section refer to alternatives of the CLM (it could be section 4.1. in this case) or in general? Alternative models have already been mentioned in section 3.1.
line 281: Is B_i an NxN matrix? Or we have a single B matrix with NxN parameters?
line 289: The actual physical interpretation of the latent variables in the RLM and CLM are also unclear and these models are also more descriptive models than models describing the actual generative process behind the data.
line 307: see Tkacik et al., 2013 J. Stat Mech for a simple and tractable way to describe low dimensional modulation of spikes in Ising models.
Section 6: Details of the fitting procedure are missing here, and it is also sometimes unclear what kind of models are fitted. e.g., what is the observation model for the RLM? Based on the figures, I assume that dimensions of the latent, x is 3 and the observations y is 5.

Fig2ad: What is the observation model for the RLM? Gaussian (GRLM) or Poisson? Does the model GLDS used on Fig 2D has the same likelihood as the RLM? In this case is the data generated by the RLM consistent with the GLDS - parametrised by A, C and Q? Or the difference between PCA, GLDS and RLM are due to the different likelihood used?
Fig2be: Is RLM the same as on Fig2ad? Here PLDS is just a special case of the RLM, so why the latter has a slightly lower performance?
Fig2cf: What is the performance of a trained PLDS model on this dataset? It is clear that PCA and GLDS use a wrong likelihood function.
line 371: Does LDS refer to GLDS here?
line 377: what is the training/testing procedure? What is predicted here? Spike counts of a single cell given the stimulus+observation of other neurons in the population? It would be useful to show the actual data and the predictions.

Figure 3a: error bars are missing. Does adding PSTH information to the other methods change their performance? Does the RLM found during training correspond to a PLDS model? Is the difference between PLDS and RLM because of the different training procedure or because RLM has a greater flexibility in its parameters?
Fig4 shows only pooled data. It would be good to show cell-by-cell comparisons of e.g., the decay or the amplitude of the CCG.

line 412: this part is a not really connected to the previous parts of the paper. I think that the procedure of training/testing is slightly different here, but there is no indication of this in the text.

I appreciate the detailed response of the authors and I think that including these modifications/clarifications will certainly improve the paper. I modified the scores accordingly. I still think that a table or graph showing the similarities and differences between these models could also be useful.
Summary: The paper is generally well written, although I found that the Results section is less clear than the presentation of the models. The work is original and it makes an interesting progress in the models of neural population data.

Submitted by Assigned_Reviewer_7

The authors describe two new models for neural population data with non-Gaussian observations, which are based on the idea that the correlations in the data are mostly driven by a low-dimensional latent variable. This is an important and timely question since a number of Gaussian models exist, which are easy to fit, but the available generalizations to non-Gaussian observations have various drawbacks. Overall, the paper is very well written, the approach is well motivated, most related models are discussed thoroughly and the results are convincing.

While I have no major concerns, I do think that there are a couple of issues that could be addressed to improve the paper.

One very related model that is not discussed is Gaussian Process Factor Analysis (Yu et al. 2009). Since it is easy and fast to fit, code is available from the authors, and it has been used in a number of experimental studies I think it would be nice to include in the performance comparison.

The paper may be easier to digest if it focused on the RLMs only and the CLMs would be addressed in a separate manuscript. Although I understand that the same idea underlies both models, the CLM isn't really a recurrent model (as the title of the paper suggests) and there are some aspects of the RLM that are not discussed in much depth (see, e.g. next paragraph).

The recurrent weights W seem to be an important parameter in the model, but the authors don't discuss how their structure affects the behavior of the model. Is the model guaranteed to be stable or are there values of W for which the system becomes unstable? If so, which are they? This should be discussed as it could be a drawback of the model.

I did not fully follow the derivation of the CLM (p.5), in particular how the second line of the unnumbered equation came about.

Finally, the caption of Fig. 3 mentions an RLM with self-couplings which seems to improve performance quite substantially, but it is neither shown nor has it been mentioned anywhere in the text before. If this leads to a substantial improvement, why is it not discussed and shown? If there is a good reason for not discussing/showing it, I suggest removing it from the caption as well since this creates only confusion.


[EDIT]
I read the authors' response. Their clarifications are useful and should be included in the revised manuscript. My comments with respect to including/excluding things remain.


Summary: A clearly written paper without major issues. The manuscript could be improved at a few places and focusing on the RLM would potentially create space for discussing some issues that are so far missing.

Submitted by Assigned_Reviewer_8

The authors incorporated classical ideas from Kalman filtering and recurrent neural networks to improve upon the latent linear dynamical system for statistical modeling of neural data with long-range temporal structures. The key idea is to allow observation to feedback to influence the estimation of the hidden states in the low dimensional dynamic process. The new formulation offers several advantages: (1) deal with Poisson noise rather than limited to Gaussian noise as in latent LDS, (2) allow exact fit and fast computation. The data modeling result is shown to be significantly better than LDS. CLM is a similar idea but works on spike counts. It beats Ising model and also allow incorporation of the observations in the modeling process. This paper is well written, but with so many acronyms and jargons, it is difficult to read at times. While the advances appear to be genuine and significant and will have an impact in this subfield of neural data analysis, this reviewer is not sure whether the model can handle more realistic data better than other methods for data other from old center-out task, and wonder if it really make a difference in real applications such as motor decoding and BCI?
Summary: RLM and CLM are two related new models, created by putting the classical idea of allowing observation feedback to influence hidden state estimation back to their latent LDS type of models, that produce state-of-the-art performance in statistical modeling of neural data. However, its advantage in neural decoding or BCI remained to be demonstrated.
Author Feedback

Author rebuttal: We thank the reviewers for their helpful comments. We will incorporate suggestions into the final manuscript, where possible. We here address in some detail the major concern we identified: it is unclear to the reviewers how good the model’s performance is compared to the alternatives presented, and whether the performance gains will generalize to other datasets. We address some further points at the end.

We thought it was important to show that the RLM is competitive with LDS-type models, which were previously shown by [6] to be excellent models of neural populations. However, we did not want to go into very detailed comparisons between all models presented with the various modules that can be added to them: self-coupling terms, PSTH terms with smoothness constraints, “instantaneous correlations”, L1 or L2 regularization of the parameters on dynamics and loading matrices etc. Each of the extra modules requires sweeps over extra regularization parameters and complicates the comparison.

Rev_7 suggests including GPFA (Yu et al, 2009) into the comparison, which would be easy to do. However reference [6] has already showed the superiority of LDS models (even with Gaussian likelihood) over GPFA on similar data.

Rev_7 also points to our comment in the caption of figure 3, that an RLM with self-couplings (SC) significantly enhances performance. This refers to a model where a refractory-like term, expressed in a set of basis functions, has been added to the predicted rate $\hat{y}_t$. For the performance comparisons we have generally tried to keep all models as simple as possible, hence by default the LDS-type models evaluated and the RLM models did not contain this self-coupling term. These models without SC should be directly compared to the GLM - SC model in figure 3 (a), where we artificially removed the self-coupling basis functions in the GLM. The full GLM model can be directly compared to the RLM+SC model: the difference between these grows to >80% of the predictive MSE of the GLM. Additional performance can be gained by using PSTH terms in the RLM (mu_t in line 176 and supplementary figure 5).

Rev_8 is not sure whether the RLM can handle other data than that used in the reported experiments. Data from centre-out reaches such as these have been widely used in the development of dynamical population models. Recently, we have also modelled array recordings from two sensory cortices: auditory cortex in gerbils and somatosensory cortex in rats. In both cases the RLM outperformed GLM-type models substantially, even when a large amount of training data was available (1024 trials of 2.5 seconds each of stimulation with identical sequences of complex sounds).

A separate question of Rev_8 is whether our work will translate to improvements in BCI or neural decoding. As a statistical model, the RLM does not directly attempt decoding and we have limited this paper to statistical modelling of the spike trains. Nonetheless, we do have separate novel results on using RLM dynamics for decoding hand position from M1 activity. Compared to a state-of-the-art decoder, RLM dynamics reduce the decoding error during movements in *half* to an average of ~6.5mm on the test set, capturing fine details of the movements like variable onsets and the shape of the speed profile.

Rev_1 notes that the physical interpretation of the latent variables in RLM and CLM is unclear, as these are both descriptive models. However, we and others have observed that these latent variables often exhibit behavioural correlates. We did not discuss these here, both for lack of space and because we thought a NIPS audience will appreciate our theoretical work more. To illustrate a result obtained using the RLM, however: we have noted some large deflections in the latent trajectories during the delay period of the center-out task which correspond to very small physical twitches of the monkey’s hand (on the order of 1 mm). These deflections we first detected in the neural trajectories using the RLM, and only later connected them to very fine physical twitches in the hand-position data (during “fixation”). We think this is a good example of how the RLM can be used for unsupervised structure discovery from neural data.

Rev_1 asks for details of the fitting procedure: it is simple gradient descent for the RLM and CLM models and expectation maximization for LDS models (with approximate E and gradient M steps for PLDS). For the simulations the dimension of the latents and observations was 5 and 100. Instantaneous correlations were not quantitatively evaluated in combination with the RLM, except for generating probabilistic samples to produce figure 4 of the supplemental material, where we used a CLM as the observation process of the RLM. Also, for all experiments we used RLM models with Poisson observations (not Gaussian). This model class does not include all PLDS models. The Gaussian RLM class does include all GLDS models as well as some more, but due to the Poisson likelihood term this relationship holds only approximately for Poisson RLM / PLDS.

Rev_1 and Rev_7 point out that the derivation of the CLM from factor analysis is not easy to follow. We will add a step-by-step derivation in the supplementary material and potentially add more of the intermediate steps in the main text.

In response to Rev_7, the stability of the RLM is easy to assess for Gaussian likelihoods: the RLM is stable iff the matrix A - WC has eigenvalues within the unit circle. For Poisson likelihoods the output nonlinearity breaks this simple condition, but in practice we do not observe instabilities on any of the datasets.